# Effect of Vitamin B Complex Treatment on Macrophages to Schwann Cells Association during Neuroinflammation after Peripheral Nerve Injury

**DOI:** 10.3390/molecules25225426

**Published:** 2020-11-19

**Authors:** Adil Ehmedah, Predrag Nedeljkovic, Sanja Dacic, Jelena Repac, Biljana Draskovic-Pavlovic, Dragana Vučević, Sanja Pekovic, Biljana Bozic Nedeljkovic

**Affiliations:** 1Institute of Physiology and Biochemistry “Ivan Djaja”, Faculty of Biology, University of Belgrade, 11000 Belgrade, Serbia; aozhe77@gmail.com (A.E.); sanjas@bio.bg.ac.rs (S.D.); jelenag@bio.bg.ac.rs (J.R.); 2Institute for Orthopedic Surgery “Banjica”, 11000 Belgrade, Serbia; nedeljkovicpredrag@gmail.com; 3Institute for Medical Research, Military Medical Academy, 11000 Belgrade, Serbia; biljadp@gmail.com (B.D.-P.); draganavucevic@yahoo.com (D.V.); 4Department of Neurobiology, Institute for Biological Research “Sinisa Stankovic”, National Institute of Republic of Serbia, University of Belgrade, 11060 Belgrade, Serbia; sanjapekovic@gmail.com

**Keywords:** peripheral nerve injury, vitamin B complex, neuroinflammation, macrophages, Schwann cells

## Abstract

Peripheral nerve injury (PNI) triggers a complex multi-cellular response involving the injured neurons, Schwann cells (SCs), and immune cells, often resulting in poor functional recovery. The aim of this study was to investigate the effects of the treatment with vitamin B (B1, B2, B3, B5, B6, and B12) complex on the interaction between macrophages and SCs during the recovery period after PNI. Transection of the motor branch of the femoral nerve followed by reconstruction by termino-terminal anastomosis was used as an experimental model. Isolated nerves from the sham (S), operated (O), and operated groups treated with the B vitamins (OT group) were used for immunofluorescence analysis. The obtained data indicated that PNI modulates interactions between macrophages and SCs in a time-dependent manner. The treatment with B vitamins complex promoted the M1-to M2-macrophage polarization and accelerated the transition from the non-myelin to myelin-forming SCs, an indicative of SCs maturation. The effect of B vitamins complex on both cell types was accompanied with an increase in macrophage/SC interactions, all of which correlated with the regeneration of the injured nerve. Clearly, the capacity of B vitamins to modulate macrophages-SCs interaction may be promising for the treatment of PNI.

## 1. Introduction

Peripheral nerve injuries (PNI) represent a considerable health burden and a far-reaching issue of the modern lifestyle, with an estimated incidence of even ~300,000 cases per year in Europe, mostly caused by increasing rates of traffic, industrial and workplace-associated traumatism [1]. Neuroinflammation induced by PNI assumes precise orchestration of interactions between different cells, primarily Schwann cells (SCs) and phagocytic-macrophages. While latter get recruited to the injury site by cytokines released from the denervated-SCs [2,3,4,5], the recruited hematogenous macrophages, together with the resident-population, release cytokines necessary for the subsequent SCs’ activation and extracellular matrix remodeling [6].

The inflammatory and reparatory roles of macrophages were proven essential for all tissues. Accordingly, it has been shown that the macrophage depletion, following PNI, triggers an impaired process of neuroregeneration, coupled with a very poor outcome [7]. Despite this, unambiguous data about the precise macrophage contribution to neuroregeneration is still missing [8], although the importance of recruited monocytes and resident macrophages in PNI-triggered neuroinflammation has been well acknowledged. Moreover, despite how distinct resident macrophage subsets have been associated to peripheral nerves in various tissues, and linked to protective and deleterious effects on neuroregeneration, a systematic analysis of macrophage type association with peripheral nerves is still lacking [9]. Macrophages exhibit remarkable plasticity and are highly heterogeneous. According to the activation state and functions, they are classified into two polarized phenotypes: “classically activated” pro-inflammatory macrophages (M1) and “alternatively activated” anti-inflammatory macrophages (M2) [10]. So far, M1-macrophages have been recognized as primary phagocytes at the PNI site, whilst M2-macrophages subsequently take over neuroreparation, whose timely activation becomes vital to circumvent the putative M1-subset neurotoxicity [11].

SCs are considered crucial in orchestrating distinct functional modalities between macrophage subsets. Firstly, the resident-SCs, experiencing hyperproliferation, reinforce the initial macrophage activity by secreting cytokines that further recruit monocytes to the injured site [12]. As a result, SCs change the phenotype to the pre-myelinating state. Re-established interactions between SCs with the tissue-macrophages restore the remyelinating-SC phenotype, which behaves indispensable during the axonal regeneration [13]. The layers of novel myelin may stimulate macrophages to finalize the on-going inflammation at the injured nerve, so that the macrophage-SCs interaction enables fine tuning of the reparatory processes, which demands clarification to a greater extent.

Injuries of the peripheral nerves result in long-term disability and conditions characterized by wide-ranging symptoms, depending on the severity and involved nerve types. The principal PNI target is axon where microsurgery represents the first therapeutic method of choice [14]. In the context of enhanced potential for neuroregeneration [15], the development of adjuvant strategies was nowadays recognized as highly needed, transforming quickly into a novel and attractive research field [16]. Hereby, B vitamins might serve as a prominent choice, due to their broad usage in regenerative medicine, as well as in treating injuries of both central and peripheral nervous system [17].

Vitamins of the B complex act as coenzymes in a plethora of enzymatic reactions, critically affecting vital cellular functions. Numerous studies clearly revealed therapeutic potential of the B complex vitamins in peripheral nerve recovery [18,19,20]. For example, the sciatic nerve injury has been associated to decreased levels of the vitamin B complex and vitamin B12 [18], indicating that the administration of these vitamins may improve the nerve regeneration process. Moreover, a positive effect of the vitamin B12 on SC proliferation and migration has been shown, including also the myelination of axons after end-to-side neurorrhaphy in rats [21]. Further on, vitamins B1, B6, and B12, display analgesic effects in experimental animal models for acute and chronic pain, upon the neuronal injury [22]. In addition, the same vitamins have been found to promote neurite outgrowth and enhance the velocity of nerve conduction in rat acrylamide-induced neuropathy [23]. In our previously published papers we have reported that treatment with vitamin B (B1, B2, B3, B5, B6, and B12) complex could improve the recovery of the motor nerve [19], and this progress was caused by effective transition from M1-proinflammatory to M2-anti-inflammatory/reparatory macrophage phenotype, followed by the inflammatory response suppression [20]. Consistently, the aim of the present study was to demonstrate whether the same B vitamins cocktail (B1, B2, B3, B5, B6, and B12) could affect the relationship between macrophages and SCs, the essential constituents of PNI-triggered processes of neuroinflammation/neuroregeneration; thus, ultimately promoting enhanced nerve repair.

## 2. Results

### 2.1. Treatment with Vitamin B Complex Altered Macrophages/Schwann Cells Interaction during the Recovery Period after PNI

To explore the effects of vitamin B complex on the spatiotemporal relationships of macrophages and SCs during the recovery period after PNI, immunohistochemical analyses were performed on the cross sections of the femoral nerve motor branch. Results of the sham control groups (S) were compared to the profiles obtained at different investigated time points (1st, 3rd, 7th, and 14th day post operation (dpo)) in the operated (O) and in the operated groups after the administration of 1, 3, 7, and 14 intraperitoneal (i.p.) injections of vitamin B complex (OT group). We used ED1 (anti-CD68) antibody as a commonly utilized marker of activated macrophages, and S100 antibody as a well-established marker of SCs [24,25,26]. Double immunofluorescence (IF) staining was performed to visualize the CD68/S100 overlapping. As we have also noticed in our recently published paper [18], only a few ED1^+^ cells were detected in the sham operated (S) group, and this number was not significantly changed during all of the investigated time points (Figure 1(I)–Figure 4(I) A, D, G, white arrow heads and Figure 6(I) A, D, G, J). Remarkably, most of the SCs, intensively stained with S100 (Figure 1(I)–Figure 4(I) A, D, yellow arrow heads), displayed morphology of myelin-forming SCs and no overlapping of ED1 and S100 immunoreactivity was detected. In the O group, the number of ED1^+^ cells was increased at the 1st dpo, where these macrophages had round and oval morphology (Figure 1(I) B, E, H, white arrow heads) resembling the M1-type. The observed macrophages were concentrated in clusters surrounding the dark spots (Figure 1(I) B, E, K, red arrow heads), consisting of SCs with very low or without S100 immunoreactivity. Additionally, strongly stained S100^+^ cells (Figure 1(I) K, yellow arrow heads) with myelinating SCs morphology were also detected, but with no overlapping of ED1 and S100 immunoreactivity. The administration of one vitamin B complex injection was not sufficient to overcome all of the effects of PNI, although the number of ED1^+^ cells was decreased (Figure 1(I) I, white arrow heads), while the morphology of S100^+^ SCs was similar to those observed in the S group (Figure 1(I) F and L, yellow arrow heads). The total number of ED1^+^/S100^+^ cells was presented in Figure 1(II), as a number of double positive cells/mm^2^ and in Figure 1(III) as the percentage of double positive cells in ED1^+^ cells population. The number of ED1^+^ macrophages that co-localized with S100^+^ SCs was negligible in the O group (10.04 ± 0.93/mm^2^ and represented only 9.86 ± 0.76% of total ED1^+^ cells), while in the OT group it was 1.5-fold higher (15.36 ± 1.14/mm^2^), but still represented a small fraction (17.72 ± 0.62%) of total ED1^+^ cells.

Interestingly, at the 3rd dpo in the O, as well as in the OT group, we have noticed ED1^+^/S100^+^ cells (yellow fluorescence) (Figure 2(I) E and F, yellow arrows). As depicted in Figure 2(I) H and I, these ED1^+^ macrophages, found in close association with SCs, had more “foamy” morphology of the M2 type. Anyhow, these ED1^+^/S100^+^ cells were more represented in the OT group. 

The quantification of ED1^+^/S100^+^ cells was presented in Figure 2(II), as a number of double positive cells/mm^2^ and in Figure 2(III) as the percentage of double positive cells in the total population of ED1^+^ cells. The number of ED1^+^ macrophages that co-localized with S100^+^ SCs in the O group was 28.29 ± 1.26/mm^2^ and represented 21.44 ± 0.91% of total ED1^+^ cells, while in the OT group, it was 32.56 ± 0.67/mm^2^, representing 29.42 ± 0.66% of total ED1^+^ cells. Besides these ED1^+^/S100^+^ cells, in both O and OT group we have noticed macrophages with “foamy” morphology that were only ED1^+^ (Figure 2(I) H and I, white arrows), and SCs that were only S100^+^ (Figure 2(I) K and L, yellow arrow heads). In addition, in the O group, some ED1^+^ macrophages of the M1-type morphology (Figure 2(I) H, white arrow heads), were still detected around SCs with weak S100 immunoreactivity (Figure 2(I) K, red arrow heads).

During the recovery period after the PNI, the most interesting interaction pattern between macrophages and SCs was observed at the 7th dpo (Figure 3). Equally, in both the O and OT group, a huge number of ED1^+^ macrophages, predominantly with the M2-like morphology and only a paucity of ED1^+^ macrophages with the M1-like morphology were detected (Figure 3(I) H and I, white arrows and white arrow heads, respectively). In the O group we found a wide spread distribution of the dark spots (Figure 3(I) B, E, K), consisting of SCs with low S100 immunoreactivity (Figure 3(I) K, red arrow heads). These spots were surrounded with many ED1^+^/S100^+^ macrophages (yellow fluorescence) (Figure 3(I) E and H, yellow arrows), characterized by the transitional morphology (between the M1- and M2-type). These ED1^+^/S100^+^ cells were counted and the corresponding quantification was presented in Figure 3(II), as a number of double positive cells/mm^2^ and in Figure 3(III) as the percentage of double positive cells in the ED1^+^ cells population. The number of ED1^+^ macrophages that co-localized with S100^+^ SCs in the O group was 24.94 ± 0.72/mm^2^ and represented 21.22 ± 0.76% of total ED1^+^ cells, while in the OT group their number was statistically higher 37.41 ± 1.21/mm^2^, comprising 29.23 ± 0.54% of the total ED1^+^ cell population. In addition, besides these ED1^+^/S100^+^ cells, we have also noticed macrophages that were only ED1^+^, some of them showing the “foamy”, M2-type-like morphology (Figure 3(I) H, white arrows), while others displaying the M1-type-like morphology (Figure 3(I) H, white arrow heads). After the administration of seven consecutive injections of the vitamin B complex, we detected increased number of ED1^+^/S100^+^ cells (Figure 3(I) F, yellow arrows), with ED1^+^ macrophages, closely associated to S100^+^ SCs, exhibiting the “foamy” M2-type-like morphology (Figure 3(I) I, yellow arrows). Importantly, these ED1^+^ macrophages were closely associated only to S100^+^ SCs with the non-myelinating morphology (Figure 3(I) L, red arrows), while no co-localization of the ED1 immunoreactivity with the S100^+^ myelinating SCs was noted (Figure 3(I) L, yellow arrow heads).

By day 14, ED1^+^ macrophages with the M2-type morphology appeared to be prevalent in the both O and OT groups (Figure 4(I) H and I, white arrows), while S100^+^ SCs were predominantly of the myelin-forming phenotype (Figure 4(I) K and L, yellow arrow heads). Interestingly, ED1^+^/S100^+^ cells (Figure 4(I) E, yellow arrows) were mostly detected in the O group, and these ED1^+^ macrophages with the M2-type morphology were associated (Figure 4(I) E, yellow arrows) to the S100^+^ non-myelinating SCs (Figure 4(I) K, red arrow) and some myelinating SCs (Figure 4(I) K, yellow arrows). In contrast, in the OT group, reduced co-localization of the ED1 and S100 immunoreactivity was detected (Figure 4(I) F), indicating that following 14 injections of the B vitamin complex the complete transition to mature, myelin-forming SCs occurred. ED1^+^/S100^+^ cells were counted and the values obtained are given in Figure 4(II), as a number of double positive cells/mm^2^ and in Figure 4(III) as the percentage of double positive cells in the total ED1^+^ cell population. The number of ED1^+^ macrophages that co-localized with S100^+^ SCs in the O group was 21.87 ± 1.38/mm^2^ and represented 31.49 ± 1.66% of total ED1^+^ cells, while in the OT group their number was statistically lower (15.44 ± 0.69/mm^2^), and the corresponding fraction in the total ED1^+^ cell population was only 18.17 ± 0.78%.

To confirm that the S100^+^ SCs, closely associated to ED1^+^ macrophages, were the non-myelinating SCs, we performed double immunofluorescence staining with growth associated protein 43 (GAP43), a well-known marker of growing axons [27], but also a marker of non-myelinating SCs [26,28,29]. In sham controls, GAP43 immunostaining was predominantly detected in large-diameter myelinated axons, at both-time points (7th and 14th dpo) (Figure 5A,D, red asterisk). Although the majority of the myelinating S100^+^ SCs, wrapping these axons, were GAP43^−^ (Figure 5A,D, yellow arrow heads), some of them were also S100^+^/GAP43^+^ (Figure 5A,D, yellow arrows). In addition, rare non-myelinating GAP43^+^ SCs were observed as well (Figure 5D, red arrows). At the 7th dpo, the nerve tissue was damaged, the major portion of myelin sheaths was degraded and axons destroyed, while most of the SCs underwent degeneration (Figure 5B, red arrow heads). Interestingly, in the OT group, beside myelin-forming SCs (Figure 5C, yellow arrow heads), seven consecutive injections of B vitamins significantly increased the number of non-myelinating S100^+^/GAP43^+^ SCs (Figure 5C, red arrows) that wrapped multiple, small-diameter, non-myelinated axons (Figure 5C, white asterisk). Similarly, a vast number of S100^+^/GAP43^+^ non-myelinating SCs ensheathing multiple small-caliber axons (Figure 5E, red arrows) together with a paucity of myelin-forming S100^+^/GAP43^+^ (Figure 5E, yellow arrows) and S100^+^/GAP43^−^ (Figure 5E, yellow arrow heads) SCs, was detected at the 14th dpo in the O group (Figure 5E, arrow head). In contrast, in the OT, as well as in the S group, myelinating, mature S100^+^/GAP43^−^ SCs were detected as a predominant cell type (Figure 5F, yellow arrow heads), with only a few S100^+^/GAP43^+^ myelinating SCs found (Figure 5F, yellow arrows). Furthermore, strong GAP43 immunostaining was detected in large-diameter myelinated axons (Figure 5F, red asterisk).

The comparative presentation of time-dependent changes in macrophages-SCs co-localization within the cross sections of the femoral nerve obtained from the S, O, and OT groups, during the investigated postoperative period (1, 3, 7, and 14 days) and upon the administration of 1, 3, 7, and 14 injections of the B vitamin complex, was depicted in Figure 6(I). As mentioned above, within the cross nerve sections of the S group, only a paucity of ED1^+^ cells was detected; their number did not undergo significant changes during all of the investigated time points; myelinating, mature SCs were detected as a predominant cell type and no overlapping of the ED1 and S100 immunoreactivity was noted (Figure 6(I) A, D, G and J).

In the O group (Figure 6(I) E, H and K), an intensive overlapping of the ED1/S100 immunoreactivity was seen at the 3rd, 7th, and 14th dpo, whereas in the OT group, the same was noticed after the administration of three and seven vitamin B complex injections (Figure 6(I) F, I). However, close ED1/S100 interactions were detected only between macrophages with the M2-like morphology and non-myelinating SCs, as well as between the macrophages with the M2-like morphology and SCs with low S100 immunoreactivity. In the both O and OT group, no interactions between mature, myelinating SCs and M2- or M1- type macrophages was detected. The temporal pattern of ED1^+^/S100^+^ participation in the ED1^+^ cells population in the femoral nerve cross sections obtained from Figure 6(II) O and Figure 6(III) OT experimental groups is given in the graphs (black bars). The fraction of ED1^+^/S100^+^ cells in the total ED1^+^ population varied over the recovery period. In the O group, the participation of ED1^+^/S100^+^ cells in the ED1^+^ cells population increased over the time post-injury reaching 31.49 ± 1.66% at the 14th dpo. At this post-injury time-point, a 3-fold increase was detected in the O group when compared to the 1st dpo. In the OT group, treatments with vitamin B complex increased the percentage of ED1^+^/S100^+^ cells, reaching 30% even after 3 injections and remained at the same level for seven days. However, the vitamin B complex treatment after 14 days reduced the number of ED1^+^/S100^+^ cells and their fraction in the total ED1^+^ cell population was only 18.17 ± 0.78%. Given that interaction between macrophages and mature, myelin-forming SCs was negligible, obtained results indicated that treatment with the B vitamin complex after 14 days triggered the complete transition to mature, myelin-forming SCs. Based on these results, it can be concluded that PNI alters interactions between macrophages and SCs in a time-dependent manner, while the treatment with the B vitamins complex accelerates the transition from the non-myelin to myelin-forming SCs type and from M1- to M2-like macrophage morphology. 

### 2.2. Administration of the Vitamin B Complex Reduced the Expression of Proinflammatory Cytokine TNF-α in SCs after the PNI

Next, we wanted to investigate how the PNI affects the expression profile of proinflammatory cytokine tumor necrosis factor alpha (TNF-α) in SCs and whether the treatment with B vitamins could modulate this TNF-α expression pattern after the PNI. 

In our model of the femoral nerve transection at the 14th dpo we have noted increased expression of TNF-α within cross sections of the operated nerve (O) (Figure 7H) compared to the sham-operated controls (S) (Figure 7G). The administration of 14 injections of the vitamin B complex (OT group) reduced the TNF-α expression (Figure 7I), which was, however, still higher compared to the S group (Figure 7G). Interestingly, besides in macrophages with the M2-like morphology (Figure 7E,F,H,I, white arrows), the TNF-α immunoreactivity was detected in some (Figure 7E,F,H,I, yellow arrows), but not all SCs (Figure 6E,F,H,I, yellow arrow heads).

### 2.3. Effect of PNI and the Vitamin B Complex Treatment on the Expression of Anti-Inflammatory Cytokine IL-10 in SCs

Further, we investigated the expression of anti-inflammatory cytokine interleukin 10 (IL-10) in SCs in all of the examined (S, O, and OT) groups. IL-10 immunoreactivity was detected in all of these groups (Figure 8G–I), being mostly pronounced in the O group (Figure 7H). Strikingly, the bulk of IL-10 immunoreactivity was noticed in IL-10^+^ cells resembling M2-macrophages, although IL-10^+^/S100^+^ cells were abundantly present as well (Figure 8E,F, yellow arrows). In the OT group, the overlapping of IL-10/S100 immunoreactivity (Figure 8F,I, yellow arrows) was detected in mature, myelinating SCs, albeit the larger part of S100^+^ SCs were IL-10^−^ (Figure 8F, yellow arrow heads). The similar pattern of IL-10 immunoreactivity was seen in the S group, but IL-10^+^/S100^+^ cells were less represented (Figure 8D,G, yellow arrows, and yellow arrow heads).

Using serial transversal sections we were able to visualize tight interactions between M2-macrophages and SCs, that were aligned to form bands of Büngner (Figure 9A–C) and were intensively stained with IL-10 (Figure 9D–F). 

## 3. Discussion

The aim of this study was to highlight the molecular mechanism underlying previously detected B vitamins-induced locomotor activity improvement after PNI. Herewith, the macrophages-SCs interaction, following the peripheral nerve controlled transection emerged as an important target of the applied treatment [19,30]. Noteworthy, these macrophages-SCs interactions were modulated in a time-dependent manner either post-PNI alone, or upon B vitamins application. However, the treatment accelerated the transition from the non-myelin- to myelin-forming-SCs phenotype. Furthermore, the stimulation of the M1-to-M2 macrophage phenotype switching consequently altered macrophages-SCs interactions.

Previously, we have shown that the vitamin B complex treatment effectively promotes PNI-induced M1-to-M2 macrophage polarization and suppresses inflammation, by reducing the expression of proinflammatory and up-regulating the expression of anti-inflammatory cytokines [20]. Herewith, we address the relationship between macrophages and SCs, the most fundamental cell-to-cell interaction during the PNI-triggered neuroinflammation. Hereby, Wallerian degeneration affects the nerve stumps distal to the lesion, which are not directly physically traumatized. SCs initiate the elimination of damaged axons by rejecting the myelin and, subsequently, recruit the bone-marrow-derived macrophages together with activated-resident-SCs for tissue debris removal [31]. Our results indicate the copious presence of destructed axon areas in the injured nerve of O animals, at the 1st dpo, concurrently with the dedifferentiation of demyelinated-SCs. Consistent to this, in young rats, such as our animals, the lag period separating the injury and axon degeneration involves the first 24–48 h [32]. The detached axon segments remain intact for days post-PNI, and can still transmit action potentials when stimulated [33,34]. Accordingly, the noted decrease in the number of M1-like-macrophages along with the preserved SCs morphology and myelination in the injured OT animals nerve, led us to hypothesize that the B vitamins treatment may prolong the lag period and reduce the extent of axon degeneration. 

After the first period of intensive PNI-induced axon destruction, at the 3rd dpo in both O and OT animals, we noticed ED1^+^ macrophages closely associated to SCs, displaying more “foamy”-M2-type morphology, particularly in the OT group. Conversely, in the O group, ED1^+^ macrophages of M1-type-morphology and SCs with low S100 immunoreactivity were still detected. According to a study [8], it was proposed that SCs most likely support the macrophage PNI-functioning via expressing several ligands known to interact with macrophage receptors, thus regulating the M1-to-M2 transition. SCs secrete classical M2-associated cytokines and behave as potent inducers of M2-macrophages. These, in turn, stimulate tissue repair, via promoting remyelination by activating endogenous SCs. Moreover, since macrophages were shown to regulate PNI-triggered SCs maturation, one could not exclude that macrophages-SCs interaction operates vice-versa as well [35].

The most interesting pattern of post-PNI macrophage-SC interaction was noted in our study at the 7th dpo. Regardless of the treatment conditions, we detected an extensive repertoire of ED1^+^ macrophages and SCs in the injured nerve. M2-like-macrophages appeared predominant, with only a small fraction of M1-like-cells observed. Additionally, in the O group, a widespread distribution of the dark spots representing damaged axons was noted marginal to SCs, with weak S100 immunoreactivity, probably undergoing degeneration/dedifferentiation. The analogous was not observed after the B vitamins treatment. Interestingly, these areas of axonal/SC-degeneration appeared borderline to many ED1^+^/S100^+^ macrophages displaying transitional M1-to-M2 morphology. Macrophages with the M1- or M2-morphology were also present. On the other hand, we noticed that B vitamins significantly increased S100^+^ SCs closely associated to ED1^+^ cells. Importantly, these ED1^+^ macrophages displayed the M2-like-morphology and were closely associated to only S100^+^ SCs of non-myelinating morphology, wrapping multiple small-diameter non-myelinated axons and being GAP43^+^. 

A similar profile of M2-macrophages to non-myelinating-SCs interaction was detected in the O nerve at the 14th dpo. These S100^+^ SCs were also GAP43^+^. Considering that GAP43, a marker of growing axons [27], may also represent a marker of the non-myelinating-SCs [26,28,29], the overlapping between S100 and GAP43 immunoreactivity classifies implicated SCs to the non-myelinating class. Contrary, only a paucity of the myelin-forming (S100^+^/GAP43^+^ and S100^+^/GAP43^−^) SCs was detected in the O group. However, after 14 days of exposure to B vitamins the myelinating, mature SCs, which were GAP43^−^, appeared as the predominant SC-type in the OT group, while a strong GAP43 immunostaining was detected in the large-diameter myelinated-axons. 

As evidenced in our study, PNI causes the destruction of the majority of SCs at the 7th dpo, as well as reprogramming from the myelin to non-myelin-forming (Remak) SCs. This aligns with both SC classes undergoing large-scale gene expression post-PNI, leading to the specialized, repair-promoting phenotype [36]. Given that the Remak SCs unsheathed uninjured fibers and are capable of acting as “sentinels” of injury/disease in proximity [37,38], it is not surprising that exactly this SCs phenotype appears most abundant at the 7th and 14th dpo.

Considering all, we can safely conclude that the B vitamins treatment protects myelin-forming SCs and accelerates the appearance of non-myelin-forming SCs. This preserves the functionality of the injured femoral nerve, as manifested by enhanced recovery of the locomotor performances in rats, which was demonstrated previously by our group [19]. Consistently, some recent publications report positive effects of individual B vitamins application (B12 > B1 > B6) on the damaged sciatic nerve repair by affecting myelination and SCs. Importantly, to obtain an optimal regenerative effect, the usage of B vitamins cocktail was proposed [39]. Moreover, beneficial effects of vitamin B12 were acknowledged in a focal demyelination rat model, in terms of accelerated re-myelination, improved recovery of motor/sensory functions, and stimulation of SCs differentiation [40]. Likewise, folic acid may stimulate the post-PNI repair by promoting SCs proliferation and migration, and secretion of nerve growth factors [41].

As outlined above, during the post-PNI recovery, we observed time-dependent changes in macrophage and SC morphology in terms of transition from the round-shaped, smaller M1-, to the “foamy”-shaped, larger M2-macrophages, and non-myelinating-SCs to the myelinating-mature-SCs. Concerning this, we conclude that the B vitamins treatment balances the macrophages-to-SCs interaction to limit the injured nerve damage by accelerating the transition from indispensable inflammation to neuroreparation right after PNI. 

Consistently, we demonstrated that PNI affects the expression profile of TNF-α and IL-10 in SCs, this being modulated through the administration of B vitamins for 14 dpo. Following the sciatic nerve transection a phasic TNF-α mRNA expression pattern was observed, peaking immediately (14 h), after 5 days and also two weeks [42,43]. In our model of the femoral nerve transection, we also noted increased expression of TNF-α at the 14th dpo within O, compared to S animals, while the administration of B vitamins reduced the TNF-α immunoreactivity. Apart in M2-macrophages, the TNF-α immunoreactivity was detected in some, but not all SCs. Interestingly, the B vitamins treatment reduces the TNF-α expression, thus, protecting myelin-forming-SCs that produce IL-10. Hence, the preserved femoral nerve functionality is manifested as increase in GAP-43 expression and improved locomotor recovery. 

IL-10, whose up-regulation was shown from 7 up to 28 days post-injury [44,45], was proposed to modulate the proinflammatory cytokines expression and axonal plasticity [46]. In two different PNI models [42,43], the expression of IL-10 mRNA underwent gradual increase during Wallerian degeneration, while the recent results [20] imply the prevalence of macrophages expressing IL-10, which represents the M2-(anti-inflammatory)-phenotype marker [47,48,49,50] at the 14th dpo. Remarkably, fractions of ED1^+^/IL-10^+^ cells in total ED1^+^ population were equivalent between O and OT animals. Moreover, some IL-10^+^ cells lacking ED1 immunoreactivity, with SCs-like morphology were observed as well [20]. In the present study, at the 14th dpo bulk of IL-10 immunoreactivity was detected in ED1^+^ M2-like-morphology macrophages, and in IL-10^+^/S100^+^ cells, which were also abundantly present. We assumed that increased IL-10 expression in “foamy”-M2-macrophages [20] together with the IL-10^+^/S100^+^ SCs presence may contribute to the resolution of PNI-triggered inflammation/nerve repair. In contrast, B vitamins treatment diminishes the overlapping of IL-10 and S100 immunoreactivity in myelinating, mature SCs, with larger fraction of S100^+^ SCs being IL-10^−^, as seen in the S group. Analogous inflammatory profile of post-PNI SCs was obtained by Dubový et al. [51], who suggested that such a simultaneous induction of proinflammatory and anti-inflammatory cytokines balances PNI-induced inflammation to promote axonal growth.

Accordingly, the main aim of our study was to investigate the association between different types of macrophages and SCs after PNI and to explore whether the treatment with the vitamin B complex could influence this relationship. We have clearly demonstrated that only M2-repair-promoting macrophages were in close-association with SCs, particularly of the non-myelinating type, while no co-localization between macrophages and myelin-forming SCs was observed. Regarding the quantification of ED1^+^/S100^+^ cells and their fraction in the total ED1^+^ cell population, these results gave us the information about the extent of ED1^+^ macrophages and S100^+^ SCs interaction during the recovery period after PNI and also how the B vitamin treatment affected the temporal profile of the corresponding interactions, all telling us about the success/extent of the recovery. Thus, in the O group, we have noticed that at the 14th dpo the prevalence of ED1^+^/S100^+^ cells in the total population of ED1^+^ cells was the highest. This suggests that interactions between macrophages and SCs are intense. Most likely, the majority of these SCs belong to the non-myelinating SCs class, while ED1^+^ macrophages belong to the M2 phenotype, being involved, together with aforementioned non-myelinating S100^+^ SCs, in the formation of Büngner bands, which are shown to represent the regeneration tracks for directing axons to their targets [19,36]. In contrast, after 14 days of the treatment with the vitamin B complex we have noted significant reduction in the number of ED1^+^/S100^+^ cells and their fraction in the total ED1^+^ cell population. Most of the axons had a renewed myelin sheath and myelin-forming SCs were the predominant type of SCs, as we have also seen in the sham-control nerve sections. Given that the interaction between macrophages and myelin-forming SCs was negligible, obtained results indicated that 14 days of the consecutive B vitamin complex treatment triggered the complete transition to mature, myelin-forming SCs that wrapped large-caliber axons intensively labeled with GAP43, a marker of axonal outgrowth. These results suggested that, by the 14th dpo, the regeneration of injured nerve and the recovery of muscle function gets completed after the treatment with B vitamins, which we have confirmed with behavioral and electromyography testing in our previously published paper [19].

Axon regeneration proceeds at a rate of 1–3 mm/day and depends on the location along the neuron, as well as cytoskeletal materials and proteins, such as actin and tubulin. Further elongation happens through the remaining endoneurial tube, which directs axons back to their original target organs. SCs are essential at this stage of regeneration, as they form Büngner repair bands, which protect and preserve the endoneurial channel. Moreover, together with macrophages, SCs release various neurotrophic factors to stimulate nerve regrowth. After reaching the endoneurial tube, the growth cone has a higher probability of reaching the target organ and triggers the maturation process. This process involves remyelination, axon expansion, and ultimately, functional re-innervation [52]. Related to this, in this study, tight M2-macrophages-to-SCs interactions were confirmed in transversal sections of the injured nerve. Moreover, we clearly demonstrated that the IL-10 immunoreactivity was associated with M2-macrophages, but also with SCs forming the Büngner’s bands. Consistent with the literature data [35], our results confirm the role of macrophages as regulators of SCs maturation after PNI.

Versatile effects of the investigated B vitamins treatment stand as an important result, since most compounds that reduce neuroinflammation safeguard the myelin-forming SCs. Importantly, our data show that following PNI, a balance in myelin-forming SCs protection, transition to non-myelin-forming SCs, and vice-versa, can be rapidly established by applying B vitamins during the early recovery period. The underlying molecular and intracellular signaling pathways pave for more thorough clarification. Overall, macrophage/SCs plasticity induced by applying adjuvant to surgery after PNI provides a basis for macrophage/SCs-centered therapeutic strategy, as an alternative repair approach. Concerning this, in the upcoming research, the exact molecular basis of macrophage-SCs interactions in response to the B vitamins, applied hereby, remain to be thoroughly examined. 

## 4. Materials and Methods 

### 4.1. Ethical Approval and Consent to Participate

The study was approved by the Ethics Review Committee for Animal Experimentation of the Military Medical Academy and Ministry of Agriculture and Environmental Protection Republic of Serbia, Veterinary Directorate No. 323-07-7363/2014-05/5.

### 4.2. Femoral Nerve Injury Rat Model

Irintchev and colleagues described the controlled transection of the peripheral nerve as a widely used model for the examination of peripheral nerve regeneration [30]. In this study, we used adult male Albino Oxford (AO) rats (15 in total), weighing between 250 and 300 g, that were randomly divided into three groups (5 per group). Animals that underwent transection of the femoral nerve motor branch with immediate reconstruction, using a technique of termino-terminal anastomosis, form the first group of so called “operated animals” (O). The second group (OT) included animals that passed the same surgical procedure but were additionally receiving vitamin B complex therapy. The “sham operated” animals (S), which also underwent the dissection of the femoral nerve motor branch but without transection, represent the third experimental group. All of the groups were additionally divided into sub-groups (four per group), based on the post-operation day that (dpo) the animals were sacrificed on (1, 3, 7, and 14 dpo). During the entire period of the study, the animals were kept under the same environmental conditions (laboratory temperature 23 ± 2 °C, humidity between 50% and 60%, 12  h/12  h light/dark cycle with lights on at 07:00 a.m., free availability of water and food). 

As anesthesia, intraperitoneal application of ketamine (50 mg/kg; Ketalar, Eczacibasi, Turkey) and xylazine (5 mg/kg; Rompun, Bayer, Turkey) was used on all animals. Following anesthesia, the animals from all investigated groups (S, O, and OT) were appropriately positioned for identification of the femoral nerve motor branch on the rat left hind paw by skin incision in the left groin and femoral region, under aseptic conditions (as previously described [19]). In all experimental groups (S, O, and OT), the motor branch was identified just before entry into the quadriceps muscle. Subsequently, animals from the O and OT groups underwent the transection of the branch, and immediate reconstruction using a 10.0 non-absorbable suture in the form of termino-terminal anastomosis, under the microscope magnification. The skin was sutured using a 4.0 absorbable suture (Peters Surgical, Paris, France). At selected time points, the animals were sacrificed by intravenous injection of a lethal dose of ketamine/xylazine. The motor branches of the femoral nerves (both reconstructed and intact contralateral) were isolated for subsequent immunofluorescence staining. All of the procedures performed in this study were based on the rules and guidelines of the EU Directive 2010/63/EU regarding the protection of animals used for experimental and other scientific purposes.

### 4.3. Protocol for Vitamin B Complex Treatment

For the investigation of vitamin B complex treatment, ampoules (2 mL) of Beviplex (Beviplex^®^, Galenika a.d. Belgrade, Serbia), each containing B1 (40 mg), B2 (4 mg), B3 (100 mg), B5 (10 mg), B6 (8 mg), and B12 (4 µg), were used. The given dose was 1.85 mL/kg/day. The complex of B vitamins was injected intraperitoneally immediately (15 min) after the operation and then every 24 h from the day of the operation until the day of sacrifice. Operated, but untreated animals (O) were intraperitoneally injected with the same volume of physiological solution. 

### 4.4. Femoral Nerve Processing Procedure for Immunofluorescence Staining

In the Laboratory for Pathohistology and Cytology HistoLab, Belgrade, all of the isolated motor branches of femoral nerves were prepared for immunohistochemistry in this study. The isolated nerve samples underwent the fixation procedure in the 10% formaldehyde solution to preserve the tissue morphology and antigenicity of target molecules on the dissected nerve. Prior to the addition of melted paraffin wax, the isolated nerves underwent a series of dehydration steps at room temperature (RT): (1) 3 × 30 min in 70% ethanol; (2) 3 × 30 min in 90% ethanol; (3) 3 × 30 min in 100% ethanol; and (4) 3 × 30 min in xylene. Following dehydration, the tissue was immerged into the melted paraffin wax at 58 °C. Microtome sectioning of the paraffin-embedded tissue was next done at a thickness of 5 µm. Sections were then incubated at 56 °C in water bath, mounted onto histological slides, pre-coated with gelatin for better tissue adhesion, and dried overnight at RT.

### 4.5. Procedure of Immunofluorescence Staining and Digital Image Processing

Immunofluorescence (IF) staining was used for protein localization on nerves slides. For indirect immunofluorescence staining, the fluorescent-dye conjugated secondary antibody, which binds to the unlabeled primary antibody, was used. The all IF staining procedures were done at RT. Only incubation with primary antibody was performed at a temperature of 4 °C. All of the solutions were prepared in 0.01 M Phosphate-Buffered Saline (PBS), pH 7.4, which was also used for washing after certain steps. Double IF staining proceeded according to the following steps:

Deparaffinization and rehydration: Microscope slides with paraffin-embedded sections were deparaffinized and rinsed in xylene 1, xylene 2, absolute alcohol, 95% alcohol, 70% alcohol, and distilled water, for 5 min in each solution. Antigen retrieval: Antigenic epitope unmasking was done by boiling microscope slides in 0.01 M sodium citrate buffer, pH 6, for 8 min at 99 °C–100 °C, followed by cooling at RT for 30 min and 3 × 5 min PBS washing. Blocking solution: after the washing step, microscope slides were incubated for 60 min in 5% blocking serum (originating from the same species as the secondary antibody) to prevent nonspecific binding of the secondary antibody. To enable membrane permeabilization, 0.5% Triton X-100 detergent was added to the blocking serum. Primary antibody, diluted in PBS, was applied onto slides and incubated overnight at 4 °C temperature. Next day, slides were washed out 3 × 5 min in PBS. Secondary antibody, diluted in PBS, was applied onto slides, where it specifically binds to the present primary antibody. Slides were next washed for 3 × 5 min in PBS. In the case of double or triple IF staining, the steps starting from the incubation in the blocking serum were repeated for the following markers. The primary and secondary antibodies used for IF labeling are indicated in the Table 1. 

After incubation with the last secondary antibody, slides were incubated in 4′,6-diamidin-2-fenilindolom (DAPI; Invitrogen, Grand Island, NY, USA) for 10 min to counterstain the nuclei and then washed 6 × 5 min in PBS and mounted with Mowiol (Calbiochem, Millipore, Germany). After drying overnight, slides were ready for viewing under the microscope. As a staining control, microscope slides that underwent the same IF procedure, but without the primary antibody application, were used.

The Carl Zeiss Axiovert fluorescent microscope, equipped with the Axiocam monochromatic camera (Axio Observer Microscope Z1, ZEISS, Gottingen, Germany), at the magnifications of 20×, 40×, and 63× was used for image processing of the prepared motor branch of femoral nerve sections and saved in .tiff format. To capture images at 63× magnification ApoTome software was used. Co-localization on the obtained fluorescent images was done using AxioVision Rel. 4.6 program, which represents a standard part of the Zeiss Axiovert microscope equipment, and then assembled and labeled in Photoshop CS6 (Adobe Systems). The quantification of single- and double-stained cells from experimental groups (S, O, OT) was performed for each time point (1, 3, 7, 14 dpo), and obtained from three independent experiments. High resolution digital images (600 pixels/inch) captured at 40× magnification (1388 µm × 1040 µm) (three images/group/independent experiment) were used for cells counting. The total number of single- or double-positive cells was counted manually using Adobe Photoshop Creative Cloud (Version 14.0). Additionally, the percentage of double-positive cells in some investigated cells populations was calculated and presented.

### 4.6. Statistical Analysis

Statistical comparison between two experimental groups was performed using a two-sided Student’s t test and a value of *p* < 0.05 or less was considered significant. Values were shown as mean values with standard error (SEM).

## 5. Conclusions

In this study we report for the first time that the treatment with the complex of B vitamins (B1, B2, B3, B5, B6, and B12) could effectively promote PNI-induced transition of the non-myelinating to myelin-forming-SCs phenotype and suppress neurodegeneration by reducing the expression of proinflammatory and up-regulating the expression of anti-inflammatory cytokines, produced by macrophages and SCs. This consequently changes interactions between these cells, thereby contributing to the regeneration of the injured nerve. In conclusion, the ability of B vitamins to modulate macrophages-SCs interaction reveals their potential as an additional tool in peripheral nerve regeneration therapies in humans, which requires extensive further research and confirmation in clinical trials.

## Figures and Tables

**Figure 1 molecules-25-05426-f001:**
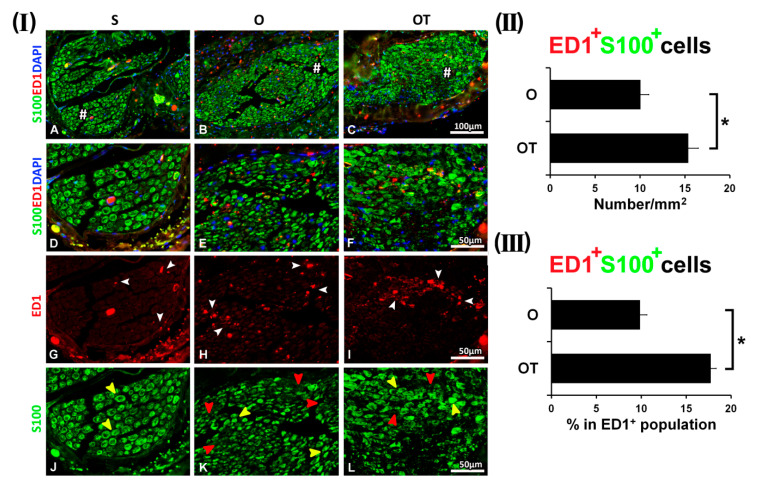
Effect of peripheral nerve injury (PNI) and treatment with B vitamins on Schwann cells (SCs)-macrophages co-localization at the 1st day post operation (dpo). (**I**) Cross sections of femoral nerve obtained from sham (S), operated (O, the transection of the motor branch and immediate reconstruction using termino-terminal anastomosis) and operated and treated with the vitamin B complex (B1, B2, B3, B5, B6, and B12) (OT) groups were stained for ED1 (anti-CD68, red) antibody, a marker of activated macrophages. Anti-S100 antibody (green) was used as a marker of SCs. The sections were counterstained with DAPI (blue) to visualize cell nuclei. The representative images demonstrated that in the S group only a few ED1^+^ cells were detected (**G**, white arrow heads), most of the SCs had morphology of myelin-forming SCs and were intensively stained with S100 (**J**, yellow arrow heads). There was negligible overlapping of ED1 and S100 immunoreactivity (**A**–**D**). In the O group, ED1^+^ macrophages with round and oval morphology of the M1 type were abundantly present (**H**, white arrow heads), particularly around the dark spots (**E**, **K**, red arrow heads), consisting of SCs with low S100 immunoreactivity. S100^+^ cells (**K**, yellow arrow heads) of myelinating SCs morphology were also seen, but without ED1 and S100 immunoreactivity overlapping (**B**, **E**). After one vitamin B complex injection, the number of ED1^+^ cells was decreased (**I**, white arrow heads). S100^+^ cells of the myelinating SCs morphology were also present (**F** and **L**, yellow arrow heads). # indicates where the high magnification micrographs were taken from. Scale bars: **A**–**C** = 100 µm, **D**–**L** = 50 µm. (**II**) The quantification of double positive ED1^+^/S100^+^ cells from the O and OT experimental groups is depicted in the graphs (black bars) as a number of double positive cells/mm^2^ and (**III**) the percentage of double positive cells in the ED1^+^ cells population. The data are shown as the mean ± SEM of three independent experiments (three images/group/independent experiment were captured). Statistical analysis was done using a two-sided Student’s *t*-test (* *p* < 0.05 vs. O group, as indicated at the graphs).

**Figure 2 molecules-25-05426-f002:**
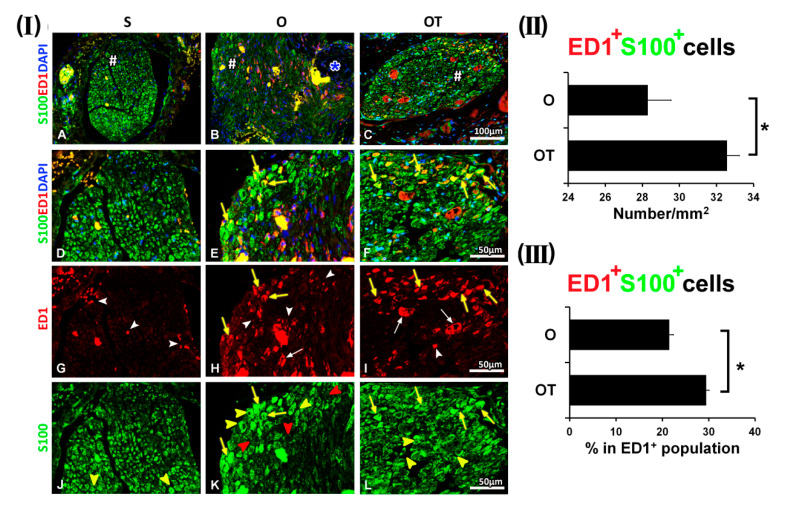
Effect of PNI and the treatment with B vitamins on SCs-macrophages co-localization at the 3rd dpo. (**I**) Cross sections of the femoral nerve obtained from the sham (S), operated (O, the transection of the motor branch) and operated and treated with the vitamin B complex (B1, B2, B3, B5, B6, and B12) (OT) groups were stained for ED1 (red), a marker of activated macrophages, anti-S100 antibody (green) as a marker of SCs, and counterstained with DAPI (blue) for visualizing cell nuclei. In the S group, a paucity of ED1^+^ cells was detected (**G**, white arrow heads), myelinating, mature SCs were the predominant cell type (**J**, yellow arrow heads) and there was no overlapping of ED1 and S100 immunoreactivity (**A**, **D**). In the O and OT groups ED1^+^/S100^+^ cells (**E** and **F**, yellow arrows) were noticed, whereby ED1^+^ macrophages, closely associated to SCs, had “foamy” morphology of M2 type (**H** and **I**, yellow arrows). In both groups, ED1^+^ macrophages with the “foamy” morphology (**H** and **I**, white arrows), and SCs that were only S100^+^ (**K** and **L**, yellow arrow heads) were noticed as well. In the O group some ED1^+^ macrophages of the M1 type morphology (**H**, white arrow heads), were detected around faintly stained SCs (**K**, red arrow heads). Blue asterisk marks the site of transection and immediate reconstruction by termino-terminal anastomosis, while # indicates where the high magnification micrographs were taken from. Scale bars: **A**–**C** = 100 µm, **D**–**L** = 50 µm. (**II**) The quantification of double positive ED1^+^/S100^+^ cells from the O and OT experimental groups is depicted in the graphs (black bars) as a number of double positive cells/mm^2^ and (**III**) the percentage of double positive cells in the ED1^+^ cells population. The data are shown as the mean ± SEM of three independent experiments (three images/group/independent experiment were captured). Statistical analysis was done using a two-sided Student’s *t*-test (* *p* < 0.05 vs. O group, as indicated at the graphs).

**Figure 3 molecules-25-05426-f003:**
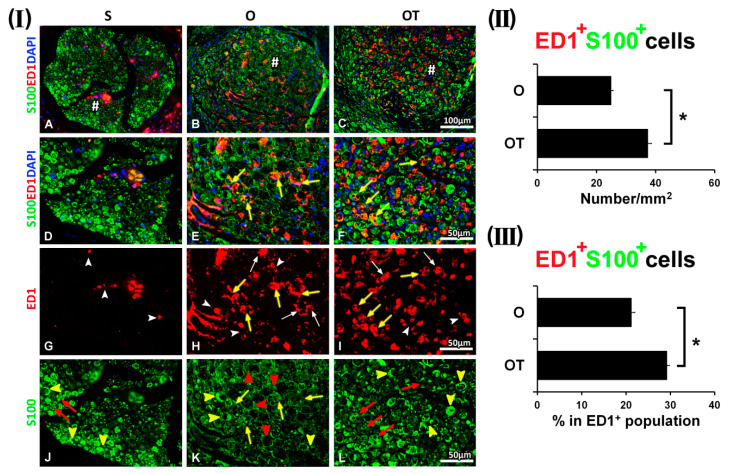
Effect of PNI and the treatment with B vitamins on SCs-macrophages co-localization at the 7th dpo. (**I**) Cross sections of femoral nerve obtained from the sham (S), operated (O, the transection of the motor branch) and operated and treated with the vitamin B complex (B1, B2, B3, B5, B6, and B12) (OT) groups were stained for ED1 (red), a marker of activated macrophages, anti-S100 antibody (green) as a marker of SCs, and counterstained with DAPI (blue) for visualizing cell nuclei. In the S group rare ED1^+^ cells were detected (**G**, white arrow heads), while besides myelinating, mature SCs (**J**, yellow arrow heads) some S100^+^ SCs of the non-myelinating morphology (**J**, red arrows) were seen. No overlapping of ED1 and S100 immunoreactivity was noted (**A**, **D**). The representative images of the O and in the OT group revealed a huge number of ED1^+^ macrophages with the predominantly M2-like morphology and a few ED1^+^ macrophages with the M1-like morphology (**H** and **I**, white arrows and white arrow heads, respectively). In the O group a widespread distribution of dark spots (**B**, **E**, **K**), consisting of faintly-stained SCs (**K**, red arrow heads) and surrounded with many ED1^+^/S100^+^ macrophages (**E** and **H**, yellow arrows), was observed. Some of the macrophages that were only ED1^+^ had “foamy” morphology of the M2 type (**H**, white arrows), while others were of the M1 type morphology (**H**, white arrow heads). Seven consecutive injections of the vitamin B complex increased the number of ED1^+^/S100^+^ cells (**F**, yellow arrows), whereby ED1^+^ macrophages of the “foamy” morphology resembling the M2 type (**I**, yellow arrows) were tightly associated to S100^+^ SCs of the non-myelinating morphology (**L**, red arrows). No co-localization of the ED1 immunoreactivity with S100^+^ myelinating SCs was noted (**F**, **L**, yellow arrow heads). # indicates where the high magnification micrographs were taken from. Scale bars: **A**–**C** = 100 µm, **D**–**L** = 50 µm. (**II**) The quantification of double positive ED1^+^/S100^+^ cells from the O and OT experimental groups is given in the graphs (black bars) as a number of double positive cells/mm^2^ and (**III**) the percentage of double positive cells in the ED1^+^ cells population. The data are shown as the mean ± SEM of three independent experiments (three images/group/independent experiment were captured). Statistical analysis was done using a two-sided Student’s *t*-test (* *p* < 0.05 vs. O group, as indicated at the graphs).

**Figure 4 molecules-25-05426-f004:**
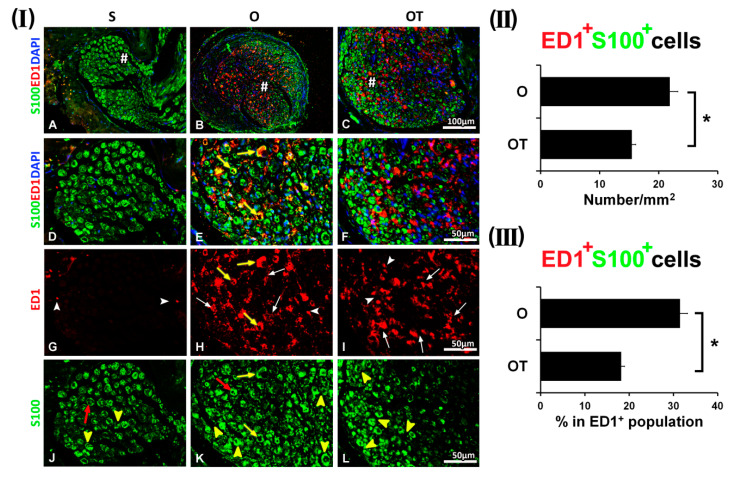
Effect of PNI and the treatment with B vitamins on SCs-macrophages co-localization at the 14th dpo. (**I**) ED1 (red), a common marker of activated macrophages, anti-S100 antibody (green), a marker of SCs and DAPI (blue) for visualizing cell nuclei, were used for immunostaining of the femoral nerve cross sections of the sham (S), operated (O, the transection of the motor branch) and operated and treated with the vitamin B complex (B1, B2, B3, B5, B6, and B12) (OT) groups. Almost no ED1^+^ cells were detected (**G**, white arrow heads) in the S group. Apart from myelinating SCs (**J**, yellow arrow heads), some S100^+^ SCs of the non-myelinating morphology (**J**, red arrows) were seen as well. There was no overlapping of ED1 and S100 immunoreactivity (**A**, **D**). ED1^+^ macrophages with the M2-type morphology were predominant in the both O and OT groups (**H** and **I**, white arrows), while S100^+^ SCs mostly belong to myelin-forming SCs (**K** and **L**, yellow arrow heads). ED1^+^/S100^+^ cells (**E**, yellow arrows) were detected only in the O group. ED1^+^ macrophages of the M2-type morphology were associated (**E**, yellow arrows) to the S100^+^ non-myelinating SCs (**K**, red arrow) and some myelinating SCs (**K**, yellow arrows). Minor overlapping of ED1 and S100 immunoreactivity (**C**, **F**) was obtained in the OT group. # indicates where the high magnification micrographs were taken from. Scale bars: **A**–**C** = 100 µm, **D**–**L** = 50 µm. (**II**) The quantification of double positive ED1^+^/S100^+^ cells from the O and OT experimental groups was depicted in the graphs (black bars) as a number of double positive cells/mm^2^ and (**III**) the percentage of double positive cells in the ED1^+^ cells population. The data are shown as the mean ± SEM of three independent experiments (three images/group/independent experiment were captured). Statistical analysis was done using a two-sided Student’s *t*-test (* *p* < 0.05 vs. O group, as indicated at the graphs).

**Figure 5 molecules-25-05426-f005:**
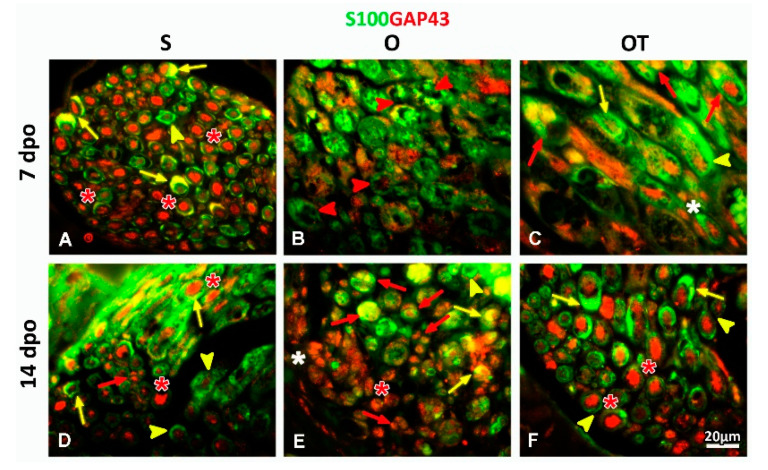
GAP43 (growth associated protein 43) expression in SCs and axons after PNI and the treatment with B vitamins. Expression of GAP43 (red) in SCs was determined in the femoral nerve cross sections obtained from the: sham (S), operated (O) and operated and treated with the vitamin B (B1, B2, B3, B5, B6, and B12) complex (OT) group at the 7th and 14th dpo. Anti-S100 antibody (green) was used as a marker of SCs. In the S group, both at the 7th (**A**) and the 14th (**D**) dpo GAP43 immunostaining was mostly detected in large-diameter myelinated axons (red asterisk), and in a few myelinated S100^+^/GAP43^+^ (yellow arrows) and non-myelinated SCs (red arrows), while S100^+^/GAP43^−^ (yellow arrow heads) SCs were predominant. (**B**) In the O group, at the 7th dpo most of the axons were destroyed and the majority of SCs degenerated (red arrow heads). (**C**) 7 consecutive injections of B vitamins (OT) increased the number of non-myelinating S100^+^/GAP43^+^ SCs (red arrows) wrapping multiple small-diameter GAP43^+^ non-myelinated axons (white asterisk). Only a few S100^+^/GAP43^+^ myelinated SCs (yellow arrows) and S100^+^/GAP43^−^ (yellow arrow heads) were seen. (**E**) At the 14th dpo, in the O group, a huge number of S100^+^/GAP43^+^ non-myelinated SCs (red arrows) unsheathing multiple small-caliber axons (white asterisk), and a paucity of myelin-forming S100^+^/GAP43^+^ (yellow arrows) and S100^+^/GAP43^−^ (yellow arrow heads) SCs was detected. (**F**) After 14 treatments with B vitamins myelinating, mature S100^+^/GAP43^−^ SCs emerged as the principal cell type (yellow arrow heads), while S100^+^/GAP43^+^ myelinating SCs were rarely present (yellow arrows). Strong GAP43 immunostaining was detected in large-diameter myelinated axons (red asterisk). Scale bar: 20 µm.

**Figure 6 molecules-25-05426-f006:**
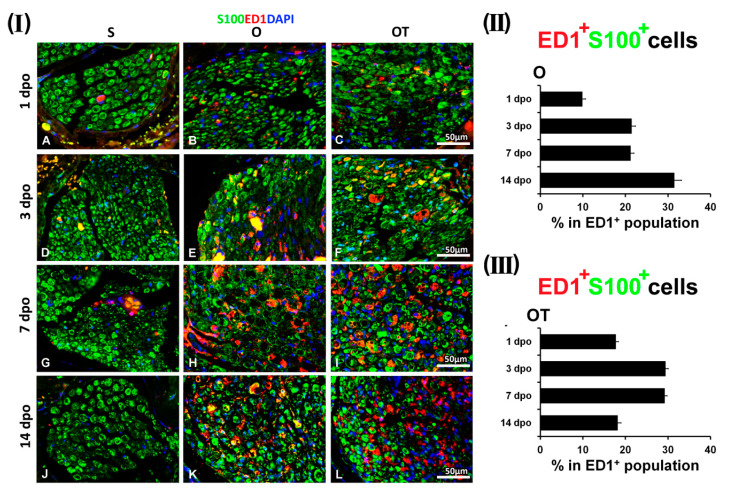
Time course immunohistochemical analysis of macrophages/SCs co-localization after PNI and the treatment with B vitamins. (**I**) Comparative analysis of macrophages/SCs crosstalk in the cross sections of the femoral nerve obtained from the sham (S) group at different time points (**A**—1 dpo; **D**—3 dpo; **G**—7 dpo; **J**—14 dpo). Time course of changes in macrophages/SCs co-localization was analyzed in the operated (O) femoral nerve during the 1, 3, 7, and 14 days of the postoperative period (**B**—1 dpo; **E**—3 dpo; **H**—7 dpo; **K**—14 dpo) and after 1, 3, 7, and 14 injections of the complex of B (B1, B2, B3, B5, B6, and B12) vitamins (OT group) (**C**—1 dpo; **F**—3 dpo; **I**—7 dpo; **L**—14 dpo). ED1 (red) was used as a common marker of activated macrophages, anti-S100 antibody (green) as a marker of SCs and DAPI (blue) for visualizing cell nuclei. Scale bar: 50 µm. Time-dependent changes in the percentage of double positive ED1^+^/S100^+^ cells in ED1^+^ cells population from the (**II**) O and (**III**) OT experimental groups was depicted in the graphs (black bars). The data are shown as the mean ± SEM of three independent experiments (three images/group/independent experiment were captured).

**Figure 7 molecules-25-05426-f007:**
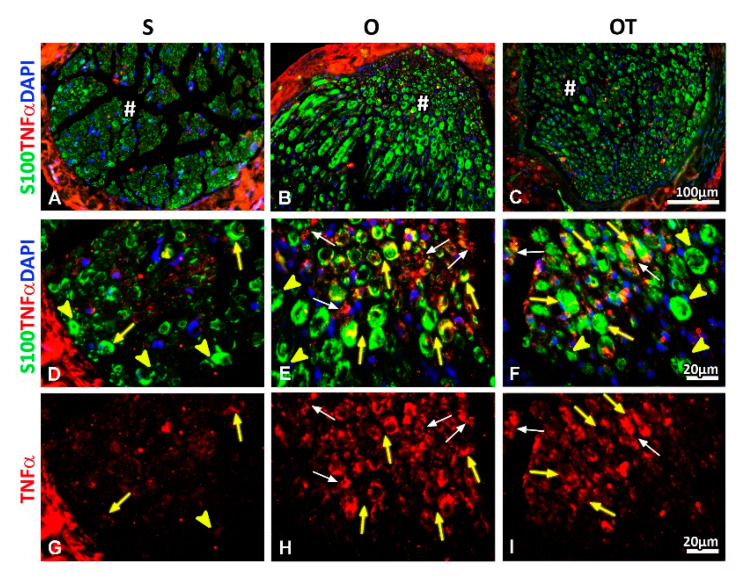
Effects of PNI and the B vitamins treatment on expression of proinflammatory cytokine tumor necrosis factor alpha (TNF-α) in SCs. Femoral nerve cross sections obtained from the: sham (S: **A**, **D**, **G**), operated (O: **B**, **E**, **H**) and operated and treated with the vitamin B (B1, B2, B3, B5, B6, and B12) complex (OT: **C**, **F**, **I**) group immunostained for TNF-α (red) demonstrated strong increase of immunofluorescence intensity in the O group (H) compared to the S group (**G**) at the 14th dpo. Immunofluorescence staining for TNF-α protein was observed in some (**E**, **F**, **H**, **I**, yellow arrows), but not all SCs (**E**, **F**, **H**, **I**, yellow arrow heads) as detected by co-localization with S100 immunostaining (green). DAPI (blue) was used for visualizing cell nuclei. Administration of 14 injections of the vitamin B complex (OT group) reduced TNF-α expression (**I**). TNF-α immunoreactivity was detected in some S100^+^ myelinated SCs (**F**, yellow arrows). In addition, TNF-α immunoreactivity was demonstrated in macrophages with the M2-like morphology (**E**, **F**, **H**, **I**, white arrows). # indicates where the high magnification micrographs were taken from. Scale bars: **A**–**C** = 100 µm, **D**–**I** = 20 µm.

**Figure 8 molecules-25-05426-f008:**
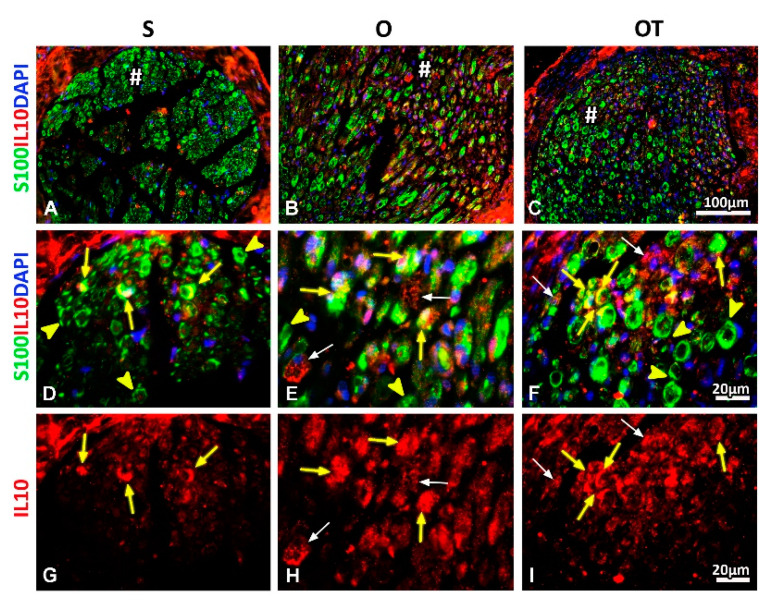
Effects of PNI and the B vitamins treatment on expression of anti-inflammatory cytokine interleukin 10 (IL-10) in SCs. Cross sections of the femoral nerve obtained from the: sham (S), operated (O) and operated and treated with the vitamin B (B1, B2, B3, B5, B6, and B12) complex (OT) group immunostained for IL-10 (red) showed strong IL-10 immunoreactivity in all of the groups (**G**, **H**, and **I**) at the 14th dpo, being the most pronounced in the O group (**H**). Increased IL-10 immunoreactivity was found in S100^+^ (green) myelinating SCs (**E** and **F**, yellow arrows) and in IL-10^+^ macrophages with the M2-like morphology (**E**, **F**, **H**, and **I**, white arrows) that were closely associated to them. In the OT group, the larger part of S100^+^ SCs was IL-10^−^ (**F**, yellow arrow heads). The similar pattern of IL-10 immunoreactivity was seen in the S group, although IL-10^+^/S100^+^ cells were less represented (**D** and **G**, yellow arrows and yellow arrow heads). DAPI (blue) was used to visualize cell nuclei. # indicates where the high magnification micrographs were taken from. Scale bars: **A**–**C** = 100 µm, **D**–**I** = 20 µm.

**Figure 9 molecules-25-05426-f009:**
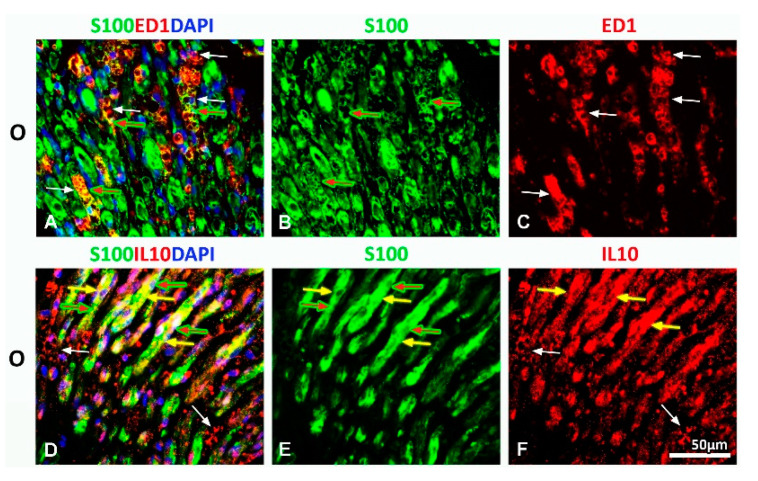
Interactions between M2 macrophages and SCs in injured femoral nerve. (**A**–**C**) We used double immunofluorescence to visualize the close contact between M2 macrophages and SCs at the 14th dpo. Serial transversal sections obtained from the operated (O) femoral nerve were immunostained for ED1 (anti-CD68, red) antibody, as a marker of activated macrophages (**C**, white arrows), anti-S100 (green) antibody as a marker of SCs (**B**, red arrows) and DAPI (blue) for visualizing cell nuclei. Complete overlapping (**A**, yellow fluorescence) of ED1 (white arrows) and S100 (red arrows) immunoreactivity confirmed tight interactions between M2 macrophages and SCs that were aligned to form bands of Büngner. (**D**–**F**) Transversal sections of the operated (O) femoral nerve immunostained with S100 (green), IL10 (red) and DAPI (blue) demonstrated that M2 macrophages (**D**, **F**, white arrows) and S100^+^ SCs (**E**, red arrows) in bands of Büngner were intensively stained with IL-10 (**D**, yellow arrows).

**Table 1 molecules-25-05426-t001:** List of primary and secondary antibodies used for immunofluorescence labeling.

Antibodies	Dilution	Company
Mouse monoclonal anti-CD68 (Clone ED1)	1:100	Abcam, Cambridge, MA, USA
Goat monoclonal anti-TNF-α	1:100	Santa Cruz Biotechnology, CA, USA
Goat monoclonal anti-IL-10	1:100	Santa Cruz Biotechnology, CA, USA
Mouse monoclonal anti-S100	1:200	Chemicon International, CA, USA
Rabbit monoclonal anti-S100	1:200	Bio-Rad Laboratories, CA, USA
Rabbit monoclonal anti-GAP43	1:200	Millipore, Darmstadt, Germany
Donkey anti-goat IgG (Alexa Fluor 555)	1:200	Invitrogen, Carlsbad, CA, USA
Donkey anti-rabbit IgG (Alexa Fluor 488)	1:200	Invitrogen, Carlsbad, CA, USA
Donkey anti-rabbit IgG (Alexa Flour 555)	1:200	Invitrogen, Carlsbad, CA, USA
Donkey anti-mouse IgG (Alexa Fluor 488)	1:200	Invitrogen, Carlsbad, CA, USA
Donkey anti-mouse IgG (Alexa Fluor 555)	1:200	Invitrogen, Carlsbad, CA, USA

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
