# Peer review of "Effect of Vitamin B Complex Treatment on Macrophages to Schwann Cells Association during Neuroinflammation after Peripheral Nerve Injury"

_molecules, 2020, doi:10.3390/molecules25225426_

Round 1
Reviewer 1 Report
The study by Ehmedah et al. explored mechanisms related to vitamin B complex on the regeneration after nerve injury. The study is well conducted and present a series of high quality immmunostaing images that illustrates the time course of B vitamin effects. It also adds data to previous publication from the group. I have only few question that I believe may improve the quality of the manuscritpt.
An English revision is recommended, as there are some typos and confuse sentences.
In the introdution, please provide more background, at least 1 paragraph, about B vitamins and nerve regeneration.
Please add information on the final dose of each vitamin received by animals and provide reference that justify the doses chosen. Additionally, compared the doses used with other studies in rats, as weel as an extrapolation for doses used in humans.
How old were the rats used, since the authors refer to young rats, but the weigh suggest adult age?
What was the lethal dose used of ketamine and xylazine?
Is it possible to provide a reference for the " Guide for the Care and Use of Laboratory Animals"?
How the quantification was performed or how the differences among groups were calculated?
Author Response
COMMENTS FROM THE REVIEWERS TO AUTHORS
At first, the authors would like to thank the Reviewers for all of their careful, constructive and insightful comments about this manuscript. All changes were highlighted in yellow in the revised version of the manuscript.
REVIEWER #1: The study by Ehmedah et al. explored mechanisms related to vitamin B complex on the regeneration after nerve injury. The study is well conducted and present a series of high quality immmunostaing images that illustrates the time course of B vitamin effects. It also adds data to previous publication from the group. I have only few question that I believe may improve the quality of the manuscritpt.
Comment_1: An English revision is recommended, as there are some typos and confuse sentences.
Response:
We revised English throughout the manuscript and made all the necessary corrections.
Comment_2: In the introdution, please provide more background, at least 1 paragraph, about B vitamins and nerve regeneration.
Response:
As the Reviewer suggested, we have now complemented the introduction with additional information regarding the effect of B vitamins on nerve regeneration (this was merged with the last paragraph, previously containing only two sentences about B vitamins). Introduced changes were highlighted yellow.
Comment_3: Please add information on the final dose of each vitamin received by animals and provide reference that justify the doses chosen. Additionally, compared the doses used with other studies in rats, as weel as an extrapolation for doses used in humans.
Response:
We thank the referee for this suggestion and provide the corresponding explanation in the following paragraph.
Applied doses of B vitamins were calculated according to the non-published results of Dacic’s doctoral thesis, exploring the effects of the vitamin B complex (B1, B2, B3, B6, and B12) on the recovery of motor function after unilateral sensorimotor cortex ablation of the rat forebrain. In this study, the following doses of B vitamins were applied: B1 (33 mg/kg/day), B2 (7.5 mg/kg/day), B3 (500 mg/kg/day), B6 (33 mg/kg/day), and B12 (0.5 mg/kg/day). The results indicated that the cocktail of B (B1, B2, B3, B6, and B12) vitamins alone, or in combination with ribavirin, could serve as an effective adjuvance to established therapeutic approaches in the treatment of brain injuries. Animals from the study of both CNS and PNS injury did not show any side effects upon the therapy application.
In the present study, during the first 14 days upon the peripheral nerve injury, we used daily administration since this early period is critical for the upcoming neuroregeneration (Scheib and Höke, 2013). Note that in the M&M section, we provided the following information: “For the investigation of vitamin B complex treatment, ampoules (2 mL) of Beviplex (Beviplex®, Galenika a.d. Belgrade, Serbia), each containing B1 (40 mg), B2 (4 mg), B3 (100 mg), B5 (10 mg), B6 (8 mg), and B12 (4 µg), were used. The given dose was 1.85 mL/kg/day.” The treatment started 15 min after the PNI and then continued for 1, 3, 7, or 14 days, every 24h. After the calculation of volume and doses that we cited in M&M, the cocktail of B vitamins was given i.p. in the following doses: 37 mg/kg/day (B1), 3.7 mg/kg/day (B2), 93 mg/kg/day (B3), 9.3 mg/kg/day (B5), 7.4 mg/kg/day (B6), and 3.7 µg/kg/day (B12).
The corresponding combination and doses from the aforementioned PhD thesis is provided in the table below.
|
Daily recommended dose in humans |
Doses applied in the model of brain injury in PhD thesis of Sanja Dacić |
Dose applied in our PNI model study |
B1 |
1.1 * |
33.0* |
37.0* |
B2 |
0.1 * |
7.5* |
3.7 * |
B3 |
2.9 * |
500.0* |
92.6 * |
B5 |
0.3+ |
- |
9.3+ |
B6 |
0.2 * |
33.0* |
7.4 * |
B12 |
0.1 + |
0.5+ |
3.7+ |
* mg/kg/day.; + µg/kg/day
Comment_4: How old were the rats used, since the authors refer to young rats, but the weigh suggest adult age?
Response:
Adult animals, about three months old were used in the study. Please, note the information provided in the M&M section “In this study we used adult male Albino Oxford (AO) rats (15 in total), weighing between 250 and 300 g, that were randomly divided into three groups (5 per group)”. As for referring to young animals, note that the cited publication used animals of the similar age, as judged by the corresponding weight provided (240-260g weighting Wistar rats). All these rats are young adults, since the term old refers to the animals 2 (and above) years old.
Comment_5: What was the lethal dose used of ketamine and xylazine?
Response:
Ketamine–xylazine represents a commonly used combination for anesthesia and euthanasia. Ketamine–xylazine is most often given intraperitoneally as an anesthetic combination. We used 200 µL of a 10:1 mixture of ketamine (100 mg/mL) and xylazine (100 mg/mL).
References:
American Veterinary Medical Association. [Internet]. 2007. AVMA guidelines on euthanasia, 2007 update, p 4–11. [Cited Jan 8 2009]. Available at http://www.avma.org/issues/animal_welfare/ euthanasia.pdf
Schoell AR, Heyde BR, Weir DE, Chiang PC, Hu Y, Tung DK. Euthanasia method for mice in rapid time-course pulmonary pharmacokinetic studies. J Am Assoc Lab Anim Sci. 2009 Sep;48(5):506-11. PMID: 19807971; PMCID: PMC2755020.
Comment_6: Is it possible to provide a reference for the "Guide for the Care and Use of Laboratory Animals"?
Response:
This was now corrected. The manuscript was complemented with appropriate reference (Guide for the care and use of laboratory animals).
Comment_7: How the quantification was performed or how the differences among groups were calculated?
Response:
The quantification for Figures 1, 2, 3, 4, and 6 was now done and the appropriate graphs were added to these figures. The quantification procedure was described in detail in the newly added M&M subsection.

Reviewer 2 Report
In the manuscript entitled “Effect of Vitamin B Complex Treatment to Crosstalk between Macrophages and Schwann cells during Neuroinflammation after Peripheral Nerve Injury” Ehmedah et al. analyse the kinetics of macrophage recruitment and activation after peripheral nerve injury. Furthermore, the authors visualise the interactions between Schwann cells and macrophages (presumably with different activation programs) and how they change as inflammation resolves and repair is initiated. Finally, they evaluate the effect of Vitamin B treatment in their model and correlate their observations with previous observations on the beneficial effect of the treatment.
Given the well-known role of macrophages during the resolution of inflammation and initiation of repair, the novelty of the observations of this study are limited. The neuro-immune axis; however, is relatively less characterised and is reassuring to confirm that macrophages have a conserved role across systems, including the nervous system.
Major points:
- The entire study is conducted using microscopy. Even though, the authors show representative images in all figures, some sort of quantification is required across the whole manuscript. This is the only way to unequivocally talk about significant changes in cell populations and their interaction with others.
- In many figures the authors use morphology to assess the activation program of macrophages. This is somewhat unconventional. Their conclusions would be more convincing if there was a graph with a quantification of TNFa and IL-10 expression by round/small vs. foamy/large CD68+
- The effects of vitamin B treatment are assessed at the cellular level, but functional data is lacking. The authors refer to previously published data, but it is essential to see whether this is reproducible in their hands.
- Across the manuscript the authors use the word “crosstalk” to describe the association of macrophages and Schwann cells, in my opinion this is incorrect. Crosstalk implies a type of molecular communication where two cell types are equipped with the expression of matching receptors and ligands to perform a function. In this study, the authors do not unravel such axis, they are merely looking at co-localisation.
Minor points
- In the abstract, the phrase “The treatment with B vitamins complex accelerates the transition from the non-myelin to myelin-forming SCs, thus regulating the SCs maturation, promoting the M1-to M2-macrophage polarization and consequently altering macrophages/SCs interactions, thereby enhancing the regeneration of the injured nerve” is misleading. It sounds like SCs maturation is what drives the macrophage phenotype switch, but this hasn’t been formally proven. In fact, it could be the transition of macrophage phenotype from pro-inflammatory to pro-repair is what drives SCs maturation?
- At the beginning of the manuscript, it is unclear why the authors introduce the concept of Wallerian degeneration. Is this what their experimental model tries to replicate? Also, why is it called Wallerian degeneration if the hallmark is neuroregeneration? This adds to the confusion please clarify.
- For an audience that is less familiar with the field of neurobiology it would be helpful if the authors could give more background on the Schwann cells phenotypes that they are looking at i.e. myelin-forming SCs vs. non-myelinating morphology. Do they represent injured vs. non injured cells or activated vs. resting and functionally how do they differ?
- It is not clear what the authors are concluding from Figure 9. What are the bands of Büngner indicating?
- In light of the authors findings that macrophages are selectively associated to non-myelinated SCs it would be interesting to discuss than the reparative function of macrophages is not only induced by cytokines like IL-4 but most likely by licensed molecular mechanisms that allow macrophages to recognise tissue injury:
- Bosurgi, L., Cao, Y.G., Cabeza-Cabrerizo, M., Tucci, A., Hughes, L.D., Kong, Y., Weinstein, J.S., Licona-Limon, P., Schmid, E.T., Pelorosso, F., Gagliani, N., Craft, J.E., Flavell, R.A., Ghosh, S., Rothlin, C.V., 2017. Macrophage function in tissue repair and remodeling requires IL-4 or IL-13 with apoptotic cells. Science 356, 1072-1076.
- Minutti, C.M., Jackson-Jones, L.H., Garcia-Fojeda, B., Knipper, J.A., Sutherland, T.E., Logan, N., Ringqvist, E., Guillamat-Prats, R., Ferenbach, D.A., Artigas, A., Stamme, C., Chroneos, Z.C., Zaiss, D.M., Casals, C., Allen, J.E., 2017. Local amplifiers of IL-4Ralpha-mediated macrophage activation promote repair in lung and liver. Science 356, 1076-1080.
- Minutti, C.M., Modak, R.V., Macdonald, F., Li, F., Smyth, D.J., Dorward, D.A., Blair, N., Husovsky, C., Muir, A., Giampazolias, E., Dobie, R., Maizels, R.M., Kendall, T.J., Griggs, D.W., Kopf, M., Henderson, N.C., Zaiss, D.M., 2019. A Macrophage-Pericyte Axis Directs Tissue Restoration via Amphiregulin-Induced Transforming Growth Factor Beta Activation. Immunity 50, 645-654 e646.
- It is advisable to check macrophage nomenclature guidelines and avoid M1 vs. M2 terminology, macrophage polarisation, etc.
Murray, P.J., Allen, J.E., Biswas, S.K., Fisher, E.A., Gilroy, D.W., Goerdt, S., Gordon, S., Hamilton, J.A., Ivashkiv, L.B., Lawrence, T., Locati, M., Mantovani, A., Martinez, F.O., Mege, J.L., Mosser, D.M., Natoli, G., Saeij, J.P., Schultze, J.L., Shirey, K.A., Sica, A., Suttles, J., Udalova, I., van Ginderachter, J.A., Vogel, S.N., Wynn, T.A., 2014. Macrophage activation and polarization: nomenclature and experimental guidelines. Immunity 41, 14-20.
Author Response
COMMENTS FROM THE REVIEWERS TO AUTHORS
At first, the authors would like to thank the Reviewers for all of their careful, constructive and insightful comments about this manuscript. All changes were highlighted in yellow in the revised version of the manuscript.
REVIEWER #2: In the manuscript entitled “Effect of Vitamin B Complex Treatment to Crosstalk between Macrophages and Schwann cells during Neuroinflammation after Peripheral Nerve Injury” Ehmedah et al. analyse the kinetics of macrophage recruitment and activation after peripheral nerve injury. Furthermore, the authors visualise the interactions between Schwann cells and macrophages (presumably with different activation programs) and how they change as inflammation resolves and repair is initiated. Finally, they evaluate the effect of Vitamin B treatment in their model and correlate their observations with previous observations on the beneficial effect of the treatment.
Given the well-known role of macrophages during the resolution of inflammation and initiation of repair, the novelty of the observations of this study are limited. The neuro-immune axis; however, is relatively less characterised and is reassuring to confirm that macrophages have a conserved role across systems, including the nervous system.
Major points:
Comment_1: The entire study is conducted using microscopy. Even though, the authors show representative images in all figures, some sort of quantification is required across the whole manuscript. This is the only way to unequivocally talk about significant changes in cell populations and their interaction with others.
Response:
The quantification for Figures 1, 2, 3, 4, and 6 was now done and the appropriate graphs were added to these figures, together with the corresponding statistical analysis. The quantification procedure was described in detail in the newly added M&M subsection. We thank the referee once again for this useful suggestion that will add to the quality of our results.
Comment_2: In many figures the authors use morphology to assess the activation program of macrophages. This is somewhat unconventional. Their conclusions would be more convincing if there was a graph with a quantification of TNFa and IL-10 expression by round/small vs. foamy/large CD68+
Response:
Please note that we already provided the quantification of TNF-a and IL-10 expression in CD68+ cells in our previous paper Ehmedah, A.; Nedeljkovic, P.; Dacic, S.; Repac, J.; Draskovic Pavlovic, B.; Vucevic, D.; Pekovic, S.; Bozic Nedeljkovic, B. Vitamin B Complex Treatment Attenuates Local Inflammation after Peripheral Nerve Injury. Molecules 2019, 24, 4615. In the present publication, however, our goal was to analyze the patterns of interaction between macrophages and Schwann cells, in relation to our previously published results on the expression of these cytokines in macrophages.
Comment_3: The effects of vitamin B treatment are assessed at the cellular level, but functional data is lacking. The authors refer to previously published data, but it is essential to see whether this is reproducible in their hands.
Response:
Please note that in our previous publication from the same project, we have clearly shown functional post-PNI improvement after the application of the vitamin B complex treatment (Nedeljković, Predrag, Zmijanjac, Dragana, Drašković-Pavlović, Biljana, Vasiljevska, Milijana, Vučević, Dragana, Božić, Biljana, and Bumbaširević, Marko. 2017. "Vitamin B complex treatment improves motor nerve regeneration and recovery of muscle function in a rodent model of peripheral nerve injury." Archives of Biological Sciences 69 (2): 361-368.)
Treatment with vitamin B complex enhanced recovery of walking function, decreased muscle atrophy and improved musculus quadriceps activity evaluated by EMG
Comment_4: Across the manuscript the authors use the word “crosstalk” to describe the association of macrophages and Schwann cells, in my opinion this is incorrect. Crosstalk implies a type of molecular communication where two cell types are equipped with the expression of matching receptors and ligands to perform a function. In this study, the authors do not unravel such axis, they are merely looking at co-localisation.
Response:
We would like to thank the referee for this useful suggestion. This was now corrected; throughout the revised version of the manuscript we now use the more appropriate term co-localization.
Minor points
Comment_5: In the abstract, the phrase “The treatment with B vitamins complex accelerates the transition from the non-myelin to myelin-forming SCs, thus regulating the SCs maturation, promoting the M1-to M2-macrophage polarization and consequently altering macrophages/SCs interactions, thereby enhancing the regeneration of the injured nerve” is misleading. It sounds like SCs maturation is what drives the macrophage phenotype switch, but this hasn’t been formally proven. In fact, it could be the transition of macrophage phenotype from pro-inflammatory to pro-repair is what drives SCs maturation?
Response:
We thank the referee for this useful suggestion. We now rephrased this part of the Abstract (highlighted in yellow) in the revised version of the manuscript for, hopefully, improved readability.
Comment_6: At the beginning of the manuscript, it is unclear why the authors introduce the concept of Wallerian degeneration. Is this what their experimental model tries to replicate? Also, why is it called Wallerian degeneration if the hallmark is neuroregeneration? This adds to the confusion please clarify.
Response:
We would like to thank the referee for this suggestion. This was corrected and we hopefully improved and clarified the points of Wallerian degeneration. Once again, we thank the referee for this useful suggestion.
Comment_7: For an audience that is less familiar with the field of neurobiology it would be helpful if the authors could give more background on the Schwann cells phenotypes that they are looking at i.e. myelin-forming SCs vs. non-myelinating morphology. Do they represent injured vs. non injured cells or activated vs. resting and functionally how do they differ?
Response:
Although we consider this point important, please note that detailed discussion on Schwann cell phenotypes is beyond the scopes of the present publication, so that adding new information would probably lower the readability of the manuscript. However, relevant references for interested readers are provided in both Introduction and Discussion.
Comment_8: It is not clear what the authors are concluding from Figure 9. What are the bands of Büngner indicating?
Response:
The role of Bungner repair bands is explained in the paragraph below, hopefully providing the referee all the relevant information.
After a traumatic injury, the axon gets separated into two segments: a proximal segment that remains in contact with the cell soma and a distal segment, separated from the neuron cell body (Sulaiman and Gordon 2013). The distal segment undergoes Wallerian degeneration, where axons and myelin disintegrate, while the endoneurium stays preserved. The process of Wallerian degeneration is important as it allows the elimination of molecules inhibitory to regeneration. Conversely, the proximal end undergoes retrograde degeneration up to the first node of Ranvier and debris clearance, after which it becomes ready for axon regeneration. At this stage, the proximal end begins the elongation through the distal nerve stumps so that growth cones can be detected. Axon regeneration proceeds at a rate of 1-3 mm/day and depends on the location along the neuron, as well as cytoskeletal materials and proteins, such as actin and tubulin (Menorca, Fussell, and Elfar 2013; Sulaiman and Gordon 2013). Further elongation happens through the remaining endoneurial tube, which directs axons back to their original target organs. Schwann cells are essential at this stage as they form Bungner repair bands that protect and preserve the endoneurial channel. Moreover, together with macrophages, Schwann cells release various neurotrophic factors that stimulate nerve regrowth. After reaching the endoneurial tube, the growth cone has a higher probability of reaching the target organ and starting the maturation process. This process involves remyelination, axon expansions, and ultimately, functional re-innervation (Menorca, Fussell, and Elfar 2013).
To make the long story short:
Büngner’s bands are regeneration tracks for directing axons to their targets (Jessen, K.; Mirsky, R. The repair Schwann cell and its function in regenerating nerves. J. Physiol. 2016, 594, 3521-3531).
Comment_9: In light of the authors findings that macrophages are selectively associated to non-myelinated SCs it would be interesting to discuss than the reparative function of macrophages is not only induced by cytokines like IL-4 but most likely by licensed molecular mechanisms that allow macrophages to recognise tissue injury:
- Bosurgi, L., Cao, Y.G., Cabeza-Cabrerizo, M., Tucci, A., Hughes, L.D., Kong, Y., Weinstein, J.S., Licona-Limon, P., Schmid, E.T., Pelorosso, F., Gagliani, N., Craft, J.E., Flavell, R.A., Ghosh, S., Rothlin, C.V., 2017. Macrophage function in tissue repair and remodeling requires IL-4 or IL-13 with apoptotic cells. Science 356, 1072-1076.
- Minutti, C.M., Jackson-Jones, L.H., Garcia-Fojeda, B., Knipper, J.A., Sutherland, T.E., Logan, N., Ringqvist, E., Guillamat-Prats, R., Ferenbach, D.A., Artigas, A., Stamme, C., Chroneos, Z.C., Zaiss, D.M., Casals, C., Allen, J.E., 2017. Local amplifiers of IL-4Ralpha-mediated macrophage activation promote repair in lung and liver. Science 356, 1076-1080.
- Minutti, C.M., Modak, R.V., Macdonald, F., Li, F., Smyth, D.J., Dorward, D.A., Blair, N., Husovsky, C., Muir, A., Giampazolias, E., Dobie, R., Maizels, R.M., Kendall, T.J., Griggs, D.W., Kopf, M., Henderson, N.C., Zaiss, D.M., 2019. A Macrophage-Pericyte Axis Directs Tissue Restoration via Amphiregulin-Induced Transforming Growth Factor Beta Activation. Immunity 50, 645-654 e646.
Response:
We would like to thank the referee for this suggestion. Please note, as mentioned in our response to the comment#2, we have already published the paper regarding the role of macrophages in post-PNI recovery, so in this publication we provided a comprehensive elaboration of the corresponding role of M2 macrophages.
Comment_10: It is advisable to check macrophage nomenclature guidelines and avoid M1 vs. M2 terminology, macrophage polarisation, etc.
Murray, P.J., Allen, J.E., Biswas, S.K., Fisher, E.A., Gilroy, D.W., Goerdt, S., Gordon, S., Hamilton, J.A., Ivashkiv, L.B., Lawrence, T., Locati, M., Mantovani, A., Martinez, F.O., Mege, J.L., Mosser, D.M., Natoli, G., Saeij, J.P., Schultze, J.L., Shirey, K.A., Sica, A., Suttles, J., Udalova, I., van Ginderachter, J.A., Vogel, S.N., Wynn, T.A., 2014. Macrophage activation and polarization: nomenclature and experimental guidelines. Immunity 41, 14-20.
Response:
Please note that the nomenclature from the manuscript represents an up-to-date, commonly used terminology.
References
Murray PJ. Macrophage Polarization. Annu Rev Physiol. 2017 Feb 10;79:541-566. doi: 10.1146/annurev-physiol-022516-034339. Epub 2016 Oct 21. PMID: 27813830.
Wang LX, Zhang SX, Wu HJ, Rong XL, Guo J. M2b macrophage polarization and its roles in diseases. J Leukoc Biol. 2019 Aug;106(2):345-358. doi: 10.1002/JLB.3RU1018-378RR. Epub 2018 Dec 21. PMID: 30576000; PMCID: PMC7379745.
Also note that, besides the markers we used, numerous other markers are also available to distinguish the M1 and M2 macrophage differentiation.

Round 2
Reviewer 2 Report
In this revised version of the manuscript “Effect of Vitamin B Complex Treatment to Crosstalk between Macrophages and Schwann cells during Neuroinflammation after Peripheral Nerve Injury”, the authors have quantified their observations in the microscope providing a more robust set of figures with more credible data.
There are nevertheless a few issues that I am still concerned about.
- When talking about published data by their own lab, authors might consider writing “we have previously shown”. This would help understand the reader why experiments were not repeated for this study -refer to previous comment 2 and 3.
- Even though the word crosstalk has been replaced across the manuscript, the title is still not reflecting what one can really conclude from the data. Perhaps “proximity” would be a more accurate word to be used in the tittle.
- In the abstract, the phrase “It also promotes the M1-to M2-macrophage polarization, consequently altering macrophages/SCs interactions, to enhance the regeneration of the injured nerve” implies that the macrophage phenotype dictates their ability to interact with SC, but again, this has not been proven. I think the following phrase would be more accurate: The treatment with B vitamins complex promoted the M1-to M2-macrophage polarization and accelerated the transition from the non-myelin to myelin-forming SCs, an indicative SCs maturation. The effect of B vitamins complex on both cell types was accompanied with an increase in macrophage/SC interactions, all of which correlated with the regeneration of the injured nerve.
- I am still not sure why the concept of Wallerian degeneration has to be introduced. It is actually used only 4 times in the article, and it is somehow not well connected with the study.
- In my previous comment 7, I didn’t suggest the authors to introduce a mini-revision of SC phenotypes, just clarify what myelin-forming SCs vs. non-myelinating SCs represent. In the study, their quantification has been used as a surrogate marker of what exactly?
- Regarding my previous comment 8, thank you very much for the answer but I think that some of this information needs to be added to the manuscript in order to be accessible for all the readers and comprehend the conclusions driven from the last figure.
- Regarding the nomenclature, I suggested the paper by Murray et al (Immunity, 2014) because, rather than being the most recent, it is the result of a discussion and agreement between the top macrophage labs around the world. Certainly, this is the one that I think we should all be sticking to.
- I noticed that the results in the abstract are written in present form. The convention is to write the results in past tense. It is worth going through the manuscript to check that this is the case.
- I spotted an A-D in figure legend 1 but there are only A, B and C panels. It is worth checking this kind of typos thoroughly before publication.
Author Response
COMMENTS FROM THE REVIEWER TO AUTHORS
At first, the authors would like to thank the Reviewers for all of their careful, constructive and insightful comments about this manuscript. All changes were highlighted in green in the novel revised version of the manuscript.
REVIEWER #2:
In this revised version of the manuscript “Effect of Vitamin B Complex Treatment to Crosstalk between Macrophages and Schwann cells during Neuroinflammation after Peripheral Nerve Injury”, the authors have quantified their observations in the microscope providing a more robust set of figures with more credible data.
There are nevertheless a few issues that I am still concerned about.
Comment_1: When talking about published data by their own lab, authors might consider writing “we have previously shown”. This would help understand the reader why experiments were not repeated for this study -refer to previous comment 2 and 3.
Response:
As the Reviewer suggested, we have now used that phrase in the revised manuscript.
Comment_2: Even though the word crosstalk has been replaced across the manuscript, the title is still not reflecting what one can really conclude from the data. Perhaps “proximity” would be a more accurate word to be used in the tittle.
Response:
Now we changed the title, instead of crosstalk we used the term association between the cells.
Comment_3: In the abstract, the phrase “It also promotes the M1-to M2-macrophage polarization, consequently altering macrophages/SCs interactions, to enhance the regeneration of the injured nerve” implies that the macrophage phenotype dictates their ability to interact with SC, but again, this has not been proven. I think the following phrase would be more accurate: The treatment with B vitamins complex promoted the M1-to M2-macrophage polarization and accelerated the transition from the non-myelin to myelin-forming SCs, an indicative SCs maturation. The effect of B vitamins complex on both cell types was accompanied with an increase in macrophage/SC interactions, all of which correlated with the regeneration of the injured nerve.
Response:
Thank you very much for the suggestion. This was now corrected.
Comment_4: I am still not sure why the concept of Wallerian degeneration has to be introduced. It is actually used only 4 times in the article, and it is somehow not well connected with the study.
Response:
Thank you very much for the suggestion. In the revised manuscript we removed mentioning Wallerian degeneration from the Introduction, we just used Wallerian degeneration as an explanation for some of our data, in light of the existing literature from the field.
Comment_5: In my previous comment 7, I didn’t suggest the authors to introduce a mini-revision of SC phenotypes, just clarify what myelin-forming SCs vs. non-myelinating SCs represent. In the study, their quantification has been used as a surrogate marker of what exactly?
Response:
The main aim of our study was to investigate the association between different types of macrophages and SCs after PNI and to explore whether the treatment with vitamin B complex could influence their relationship. We have demonstrated that only M2-repair-promoting macrophages were in close-association with SCs, particularly of non-myelinating type, while no co-localization between macrophages and myelin-forming SCs was observed. Regarding the quantification of ED1+/S100+ cells and their fraction in the total ED1+ cell population, these results gave us the information about the extent of ED1+macrophages and S100+ SCs interaction during the recovery period after PNI and how B vitamin treatment affected the temporal profile of their interactions, all telling us about the success of recovery. Thus, in the O group we have noticed that at the 14th dpo the prevalence of ED1+/S100+ cells in the total population of ED1+ cells was the highest. This data suggested that these interactions between macrophages and SCs are intense and that probably most of the SCs are non-myelinating SCs (as it is shown in Figure 5 the most of S100+/GAP43+ SCs have the morphology of non-myelinating SCs), while ED1+ macrophages belong to the M2 phenotype (as seen in Figure 9, ED1+ macrophages are IL10+ which confirms that they are M2 macrophages) and that together with aforementioned non-myelinating S100+ SC are involved in the formation of Büngner bands, which represent regeneration tracks for directing axons to their targets. In contrast, after 14 days of treatment with vitamin B complex we have noted significant reduction in the number of ED1+/S100+ cells and their fraction in the total ED1+ cell population. Most of the axons have a renewed myelin sheath and myelin-forming SCs were the predominant type of SCs, as we have also seen in the sham-control nerve sections. Given that the interaction between macrophages and myelin-forming SCs was negligible, obtained results indicated that 14 days of the B vitamin complex treatment triggered the complete transition to mature, myelin-forming SCs that wrapped large-caliber axons intensively labeled with GAP43, a marker of axonal outgrowth. These results suggested that by the 14th dpo the regeneration of injured nerve and the recovery of muscle function is completed after the treatment with B vitamins, which we have confirmed with behavioral and EMG testing in our previously published paper (Nedeljković, Predrag, Zmijanjac, Dragana, Drašković-Pavlović, Biljana, Vasiljevska, Milijana, Vučević, Dragana, Božić, Biljana, and Bumbaširević, Marko. 2017. "Vitamin B complex treatment improves motor nerve regeneration and recovery of muscle function in a rodent model of peripheral nerve injury." Archives of Biological Sciences 69 (2): 361-368.)
For clarity, we also incorporated this explanation in the Discussion of the revised version of our manuscript.
Comment_6: Regarding my previous comment 8, thank you very much for the answer but I think that some of this information needs to be added to the manuscript in order to be accessible for all the readers and comprehend the conclusions driven from the last figure.
Response:
Thank you very much for the suggestion. In the revised manuscript we included all the information needed, according to your suggestion.
Comment_7: Regarding the nomenclature, I suggested the paper by Murray et al (Immunity, 2014) because, rather than being the most recent, it is the result of a discussion and agreement between the top macrophage labs around the world. Certainly, this is the one that I think we should all be sticking to.
Response:
After carefully reading the suggested publication, we fully agree with the referee on the importance of setting (and following) clear standards in the field in terms of experimentation, to hopefully result with the consensus on nomenclature to use. However, this demanding goal is still to be attained and, regarding our manuscript, we found no conflicting information with the usage of either the term polarization or M1/M2 macrophages. We agree that the ability to define a spectrum of different M1- or M2- subphenotypes would be more appealing, however, our results are clearly indicative on these two main phenotypes (in terms of both cell markers and morphology) so we are forced to stick to this terminology. Nevertheless, we now referred to the suggested publication in the revised version of the manuscript since this paper truly provides a number of important guidelines and recommendations.
Comment_8: I noticed that the results in the abstract are written in present form. The convention is to write the results in past tense. It is worth going through the manuscript to check that this is the case.
Response:
Thank you very much for the suggestion. This was now corrected.
Comment_9: I spotted an A-D in figure legend 1 but there are only A, B and C panels. It is worth checking this kind of typos thoroughly before publication.
Response:
Thank you very much for the suggestion. This was now corrected.
Finally, we hope that all the referee comments are appropriately addressed both by the responses and corresponding changes in the revised version of the manuscript.
